# The Utility of Length of Mining Service and Latency in Predicting Silicosis among Claimants to a Compensation Trust

**DOI:** 10.3390/ijerph19063562

**Published:** 2022-03-17

**Authors:** Haidee Williams, Rodney Ehrlich, Stephen Barker, Sophia Kisting-Cairncross, Muzimkhulu Zungu, Annalee Yassi

**Affiliations:** 1Division of Occupational Medicine, School of Public Health and Family Medicine, University of Cape Town, Cape Town 7925, South Africa; rodney.ehrlich@uct.ac.za (R.E.); sophia.kisting-cairncross@uct.ac.za (S.K.-C.); 2School of Population and Public Health, University of British Columbia, Vancouver, BC V6T 1Z3, Canada; stephen.barker@ubc.ca (S.B.); annalee.yassi@ubc.ca (A.Y.); 3School of Health Systems and Public Health, University of Pretoria, Pretoria 0002, South Africa; muzimkhuluz@nioh.ac.za; 4National Institute for Occupational Health, Division of the National Health Laboratory Service, Johannesburg 2000, South Africa

**Keywords:** silicosis, length of service, latency, compensation, South Africa

## Abstract

In the wake of a large burden of silicosis and tuberculosis among ex-miners from the South African gold mining industry, several programmes have been engaged in examining and compensating those at risk of these diseases. Availability of a database from one such programme, the Q(h)ubeka Trust, provided an opportunity to examine the accuracy of length of service in predicting compensable silicosis, and the concordance between self-reported employment and that officially recorded. Compensable silicosis was determined by expert panels, with ILO profusion ≥1/0 as the threshold for compensability. Age, officially recorded and self-reported years of service, and years since first and last service of 3146 claimants for compensable silicosis were analysed. Self-reported and recorded service were moderately correlated (R = 0.66, 95% confidence interval 0.64–0.68), with a Bland–Altman plot showing no systematic bias. There was reasonably high agreement with 75% of the differences being less than two years. Logistic regression and receiver operating characteristic curve analysis were used to test prediction of compensable silicosis. There was little predictive difference between length of service on its own and a model adjusting for length of service, age, and years since last exposure. Predictive accuracy was moderate, with significant potential misclassification. Twenty percent of claimants with compensable silicosis had a length of service <10 years; in almost all these claims, the interval between last exposure and the claim was 10 years or more. In conclusion, self-reported service length in the absence of an official service record could be accepted in claims with compatible clinical findings. Length of service offers, at best, moderate predictive capability for silicosis. Relatively short service compensable silicosis, when combined with at least 10 years since last exposure, was not uncommon.

## 1. Introduction

The South African gold mining industry is shrinking, with approximately 95,130 people currently employed, compared with the gold mining heyday of the 1980s when the industry employed over 500,000 persons [1,2,3]. Ex-goldminers carry a substantial burden of occupational lung disease, notably silicosis and tuberculosis (TB), externalised from the gold mines to the rural areas of South Africa and neighbouring countries [4]. These include Lesotho and Mozambique, which are among the poorest countries in the world. The prevalence of radiological silicosis in this population of ex-goldminers ranges from 26 to 46% [5,6,7], while TB rates have been among the highest in any population worldwide [7,8]. Combined disease, i.e., silicosis and active TB or chest-ray (CXR) abnormalities due to prior TB, is common. 

As a result, compensation for miners’ lung disease in Southern Africa takes on a singularly important role in relieving the financial strain of premature job loss and post-employment disability. Statutory compensation for miners in South Africa falls under the Occupational Diseases in Mines and Works Act (ODMWA) and is distinct from that for general workers’ compensation [9]. Black ex-miners in particular have been poorly served by a system which was strongly racialised, with black miners receiving lesser financial benefits than white miners, in addition to suffering inferior access to benefit medical examinations [3,9,10,11,12,13]. Although there have been recent efforts to reform the statutory compensation system, difficulties of access persist, in addition to large backlogs in the processing of claims and payment of eligible claimants [14,15].

Against this background, a legal watershed was the confirmation in 2011 by the Constitutional Court of South Africa of the right of miners to pursue civil action for occupational lung disease outside the statutory workers’ compensation system [16]. This case opened the way for civil suits for silicosis against gold mining companies. The first was settled in 2013 against a former gold mining company on behalf of 23 gold miners [17] and the second in 2016 against two gold mining companies on behalf of 4365 ex-employees [18]. The latter led to the establishment of the Q(h)ubeka Trust (QT), a closed trust with the objective of providing compensation for silicosis [18,19].

The above settlements were followed by a class action suit for silicosis and, notably in addition TB, against six gold mining companies [20,21]. This was settled in 2018 with the creation of the Tshiamiso Trust, with benefits applicable to those who worked at listed mines and who contracted silicosis or work-related TB [20,21]. Unlike the Q(h)ubeka Trust, there is no limit to the number of potentially eligible claimants against the Tshiamiso Trust, which is expected to run until 2030.

In efforts to increase the efficiency and equity of administration of these statutory and civil programmes, the question has arisen about the extent to which age or occupational history information would help in identifying former workers or claims for silicosis most likely to be eligible for compensation. Age has been used on equity grounds for prioritising ex-miners for examination because of the cumulative burden of ill-health in older miners and their reduced life expectancy [22]. With regard to occupational history, there are three points at which such information could be used. The first is in invitation of former workers to compensation benefit examinations. The second is in making the clinical diagnosis of silicosis, which is based on occupational exposure history and a compatible CXR. The third is in “triaging” claims to different medical adjudication panels based on likelihood of having silicosis [15]. 

The vast majority of this population worked underground and were exposed to silica dust. In the absence of usable information on individual dust exposure or specific occupation, the most readily available occupational history metrics are years of mining service and latency.

Length of service is the most common metric used in surveillance and clinical practice. Depending on the intensity of exposure, silicosis can appear on the CXR anywhere from a few months to decades after exposure. Various thresholds of exposure for the first appearance of “chronic” or “simple” silicosis have been suggested in the literature by different authors: e.g., 10 years [23,24], 15 years [25], and even 20 years [26]. Silicosis appearing earlier than 10 years of exposure, and associated with a more severe clinical course and poorer prognosis, has been termed “accelerated” [24,27]. Given this typology, an understanding of the frequency with which claims with apparently short mining service, particularly less than 5 or 10 years of exposure, are adjudicated as compensable silicosis, would also be useful to examining medical practitioners and adjudicators.

By definition, ex-miners accumulate time after employment with retained silica in their lungs [28]. In such individuals, length of service may therefore not be a sufficient metric of silicosis risk. Further, since much silicosis is subradiological on plain chest radiography [29] and silicosis is a progressive disease [28], radiological silicosis may appear for the first time after the miner has left the industry. The term “latency”, frequently used to refer to this combination of exposed and unexposed time is, however, ambiguous. We therefore distinguish “years from first exposure”, which incorporates length of service, gaps between working contracts, and post-service time, from “years from last exposure”, which refers only to post-service time.

The difficulties facing ex-gold miners in accessing occupational lung disease compensation have been extensively documented [9,10,11,30,31]. An important barrier is that in order to be eligible for compensation, applicants have to prove their mining service, in many cases from decades earlier. The records of the primary recruitment agency for the gold mines, TEBA Ltd. (Johannesburg, South Africa), are the main source of such service data. Records were stored in hard copy until the 1970s, with subsequent electronic capture and presumably more efficient retrieval, complete only from 1983 [32]. Moreover, particularly during the 1990s, contracts of miners who were recruited directly by the mine or via labour brokers did not have their contracts captured in the TEBA Ltd. database [32]. Although a number of miners have retained some personal employment documentation, many have not [9]. However, self-reported mining service is currently not acceptable as proof of mining employment for claims purposes by either the statutory authority or the compensation trusts. 

For purposes of this study, the availability of the database of the Q(h)ubeka Trust provided an opportunity to answer questions concerning the utility of occupational information in the examination and compensation of this ex-miner population. The research objectives were two-fold:To test the agreement between self-reported and official records of length of gold mining service among these former miners;To examine distribution of length of mining service, time since first exposure, and time since last exposure among claimants certified with silicosis, and to measure the predictive accuracy of these metrics for compensable silicosis.

## 2. Materials and Methods

The Q(h)ubeka Trust was registered with the Master of the High Court of South Africa on 22 April 2016. Qualifying claimants had to lodge a claim with the Trust within three years [19]. A qualifying claimant was defined by the Trust deed as one with qualifying service (minimum of two years documented employment at specific mines in aggregate) and who was confirmed as such during the three-year claims period [19]. Potential claimants were invited to an examination for purposes of determining eligibility for compensation between April 2016 and April 2019. These examinations were performed by a network of approved medical practitioners around South Africa and neighbouring countries, and consisted of a medical history (based on completion of a questionnaire issued by the Trust), clinical examination, CXR, and spirometry. The examination and transport costs were covered by the Trust [33]. Approximately 9% of the listed 4365 claimants could not be located nor were otherwise able to lodge a claim before the claims period termination on 22 April 2019 [33].

By Q(h)ubeka Trust rules, a qualifying claimant was eligible for compensation if they were diagnosed with silicosis, with at least a 1/0 profusion on the CXR using the International Labour Organization (ILO) scale [34]. Co-existing signs of TB on the CXR (whether prior or active), progressive massive fibrosis (PMF), and/or lung function loss were taken into account for assessment of severity and grade of compensation. However, grade of silicosis was not used in the analysis in this report. 

Claims adjudication of silicosis was distributed among three medical panels each consisting of a radiologist and an experienced occupational or public health medicine specialist. Compensable silicosis is defined in this analysis as that which met clinical—including radiological—criteria for silicosis. To be paid compensation, such claimants had, however, to meet employment and other qualifying criteria defined by the Trust. For purposes of achieving a high sensitivity for silicosis, through additional scrutiny of borderline classifications, all claims in which the CXR lung profusion of silicotic opacities was read as ILO 0/1 were re-read by another panel. Separately, a separate review panel consisting of two occupational medicine specialists and a radiologist reviewed claims on application or if triggered by operating procedure. A total of 165 claims with an initial reading of <ILO 1/0 were re-classified as compensable after re-reading of the CXR by the second panel or on claim review. 

Claimant information, including age, length of service contracts, and results of the medical assessment, was captured in an information management system. This was populated through software programming that identified and extracted contract data, e.g., “start date” and “end date”, from TEBA Ltd. and mining company human resources records, and manually populated for diagnosis and CXR findings. Claimant job title was also populated from the TEBA or human resource records. 

The dataset provided two different values for length of mining service. The first was a calculation of the years worked based on the start and end dates of official records of contracts of service. The second was the number of years reported independently by the claimant and recorded in the Q(h)ubeka Trust medical history questionnaire by the examining doctor. Self-reported years of service included all mining history and not only the time spent at qualifying mines. 

Data of 3841 claimants with medical certification as of 2 November 2020 were made available to the researchers by the Q(h)ubeka Trust, in accordance with the Trust’s research policy. A subset of 695 certified claimants did not have CXR readings recorded and were excluded from the analysis; all these claimants were deceased. 

### Statistical Methods 

Statistical analyses were performed using R (version 4.0.3). Summary values were calculated using means and medians for continuous variables and frequencies for categorical variables. Pearson’s correlation coefficient was calculated to display the linear association of the two numeric metrics. Bland–Altman analysis was further used to explore limits of agreement and systematic bias between recorded and self-reported years of service [35,36]. 

Unadjusted and adjusted logistic regression analysis was undertaken with compensable silicosis as the outcome and age, length of service, years since first exposure, and years since last exposure as the independent variables. For the agreement analysis, receiver operating characteristic (ROC) analysis was employed and the area under the curve (AUC) calculated to identify the most accurate predictor of compensable silicosis [36]. Youden’s index was used to derive the point on the curve that maximised the combination of sensitivity and specificity [sensitivity + (1 − specificity)] [37]. The threshold was then varied in both directions to explore the effect of achieving greater sensitivity, or alternatively, greater specificity. 

## 3. Results

The database used contained 3146 records. Median claimant age was 66 years (interquartile range (IQR) 61–72), and median length of service on the official record was 15.1 years (IQR 8.3–20.8). Table 1 compares those claimants found to have compensable silicosis with those having non-compensable silicosis. A total of 1736 (55.2%) claims for silicosis were found compensable. Of the compensable claims, 348 (20%) had service length <10 years and 151 (8.7%) <5 years. 

A substantial number (>90) of different job titles were assigned as claimant occupation, with many appearing to involve similar tasks. It was not possible to reduce this number to valid exposure categories for analysis. 

### 3.1. Calculated vs. Self-Reported Length of Service

For purposes of the comparison of official with self-reported service, a number of records were incomplete. Of 3146 included claims, 599 (19%) had only the officially recorded value, 135 (4%) only the self-reported value, and 33 (1%) neither, leaving 2379 (76%) with both. In this subsample, there was a moderate linear correlation of 0.66 (95% CI 0.64, 0.68). The mean difference between the two metrics was 0.01 years (95% CI −0.29, 0.32) and the median difference 0.5 years (IQR −1.8, 2.1, 95% CI 0.45, 0.59) indicating no systematic bias in the form of mean over or understatement of service. Of note was that the 168 claimants with no official record of service were significantly older than those with such a record (mean 72.3 vs. 66.2 years respectively, *p* < 0.001).

Figure 1 is the Bland–Altman scatter plot of the mean against the absolute difference of the two metrics. The mean difference is not significantly offset from zero, as noted above. The majority of differences (74.2%, 1766/2379) are less than two years in either direction. Further, 89.9% (2139/2379) of the differences lie within one standard deviation of the mean (±7.5 years). Finally, the distribution of data points around the zero-difference line indicates that the lack of systematic bias is consistent across the whole range of length of service.

### 3.2. Prediction

Table 2 displays the correlation between the covariates. Of note is that there is a strong negative correlation between years worked and time from last exposure (illustrated further in Figure 2). Length of service shows a weak correlation with age, but a moderate positive correlation with years from first exposure. 

Table 3 presents the regression outcomes. All the covariates in the unadjusted and adjusted analysis had estimates of effect with 95% confidence limits excluding the null. Length of service had a stronger association with compensable silicosis than did age and time from first exposure. Time from last exposure had an inverse association, i.e., the greater the number of years since last exposure, the lower the odds of having compensable silicosis. Adjustment for length of service in model B reduced but did not eliminate this particular effect.

We also explored the relation between years since last exposure and compensable silicosis among the 887 claimants with “short” mining service, defined as <10 years (Table 4). There were only 11 of these claimants with time from last exposure <10 years, of whom one had a compensable claim. Among the claimants with >10 years since last exposure, there was no clear trend in the compensable proportion, which varied from approximately 32% to 54% with overlapping confidence intervals. A similar pattern was apparent when “very short” service (<5 years) was used as the threshold (N = 328), although with wider confidence intervals (not shown).

Table 5 provides the AUC for each metric individually and those for the adjusted models. Appendix A (see Appendix A) displays the ROC curves for unadjusted length of service and adjusted model B. These two predictors (variable and model respectively) had the highest AUCs: 0.681 for years worked, and 0.690 for the model containing age, years worked, and time since last exposure (or alternatively time since first exposure). The combined models thus contributed no additional predictive utility to that of length of service alone. We examined the effect of using self-reported rather than recorded service in the above analysis. This produced an unadjusted model (length of service) of similar predictive utility as [AUC 0.679 (95% CI 0.660, 0.700), R^2^ = 0.121].

Table 6 shows the effect of choosing different thresholds on the ROC curve. Sensitivity and specificity were maximised at a length of service of 15.7 years (Youden’s Index), both resulting in high proportions of predictive misclassification. The other thresholds showed substantial loss of specificity, or alternatively sensitivity, which changes in the complementary metric. For example, to achieve a sensitivity of 90% (predicting 10% of true compensable cases as negative), a very low length of service threshold of 6.5 years would be needed. This would result in a specificity of 27.1% (predicting 72.9% of true non-compensable claims as positive). Conversely, to achieve a 90% specificity (small proportion of “false positive” predictions), a high threshold of 23.5 years would be needed, resulting in a very low sensitivity of 25.3%.

Because length of service is more likely to be taken into account by adjudicators faced with borderline silicosis claims, we repeated the analysis excluding the 165 claims which were subject to re-reading or review under Trust procedures. This re-analysis had no effect on the overall findings.

## 4. Discussion

The aims of this study were two-fold: to determine whether there was a significant bias in self-reported length of service compared to that officially recorded; and to assess whether service-related metrics were “sufficiently” predictive to prioritise potential claimants for benefit examinations or for triage in the claims adjudication process [16]. The latter aim also enabled us to focus attention on silicosis adjudicated as compensable in claimants with apparently short service length of <10 years.

### 4.1. Concordance between Recorded and Self-Reported Service

This is the first study of the South African mining population to our knowledge to be able to compare number of years worked as reported by ex-miners to that recorded through the major recruitment agency and/or company human resource records. Where such differences do arise, recall lapses by ex-miners in both directions tend to be assumed as the explanation for the differences observed. However, when the self-reported period of service is greater than the official record, it could instead be due to lack of recording of some service periods on the recruitment company database. Lack of any official service record could be due to bypassing of the official employment register by recruitment via labour brokers or signing on directly “at the mine gate”, particularly during the 1990s [32]. Conversely, where the recorded period is the greater, this may be due to errors in official data capture, for example, service period duplication. 

Two findings from these data are notable. The first is that there is no systematic bias, with a mean difference close to zero and no tendency for bias to be related to shorter or longer length of service. The second is that there is reasonably high agreement in that three quarters of the ex-miners’ self-report a length of service that falls within 2 years of the official record. These findings should be taken into account in the clinical examination and in claims adjudication when an official record of length of service exposure is lacking. A combination of diagnosis of silicosis, self-reported service history, and a plausible history of the mode and place of recruitment may reduce the probability of an illegitimate claim to sufficiently low levels to be accepted for compensation purposes. This would benefit particularly older mineworkers. 

### 4.2. Prediction

Of the predictor variables tested, years of service had the strongest association with compensable silicosis. As noted in a previous analysis of an active gold miner cohort, job title was not a usable proxy for exposure intensity [38]. Adjusting for age and years since first or last exposure did not result in a significantly more predictive model. Either way, overall agreement (area under the curve) was only “moderate”, as were both sensitivity and specificity at the maximising threshold. Since the purpose of this project was to develop a system to identify claims of workers having a greater likelihood of silicosis for medical assessment or triage, a prediction threshold with a high sensitivity and negative predictive value would be needed in the first instance. However, since specificity declines with a rise in sensitivity, a high sensitivity would result in a substantial inefficiency in “false positives” being prioritised for assessment or referral to appropriate adjudication panels [15]. 

There is broad agreement in the literature that, depending on exposure intensity and rate, the category of silicosis designated “chronic” takes 10–15 years to appear on the CXR. In this series, claims with <10 years of exposure made up approximately 20% of certified claims based on an ILO threshold of ≥1/0. However, since this analysis is effectively a cross-sectional study of “elderly” miners, with all diagnoses made during the three-year period of the Trust’s operation, it cannot distinguish early silicosis appearing while still exposed and late silicosis occurring after exposure has ended. Whatever the case, almost all the successful claimants with short service had time since last exposure of at least 10 years, suggesting the existence of a threshold around that level. 

### 4.3. Strengths and Limitations

The strengths of the study are the large sample size, the standardised system of expert adjudication of silicosis over a defined period according to a single set of rules, and a process of quality control of adjudication decisions.

By design, the study sample consisted of invited and responding claimants in a litigation process. The sample is not representative of the population of ex-gold miners in Southern Africa and was not intended to yield population-based epidemiological measures of occurrence. This constraint should not affect the internal validity of the study in which utility of length of service and other information for a specific compensation process were the target. However, generalisability to compensation systems having different examination rules, triage processes, and definitions of compensable silicosis would be limited by the extent of such differences. 

A subset of 695 certified deceased claimants were excluded from the analysis as they did not have CXR readings recorded. It is possible that longer service workers with silicosis were excluded by death, potentially resulting in an underestimated association between years of service and the diagnosis of compensable silicosis. 

In the absence of dust exposure information on claimants, the metrics of exposure were limited to time employed and time between end of employment and claim date. Finally, while a specific length of service was not a Trust criterion for compensable silicosis, it is likely that adjudicators considered length of service in adjudicating borderline CXRs. This may have overstated the association between length of service and certified silicosis, and perhaps reduced the number of short-service silicosis claims that were accepted. To test such a bias, we excluded from the prediction analysis a small set of borderline CXR claims which were re-assigned on review, - we found no effect on the overall findings.

## 5. Conclusions

By 20 December 2021, the Q(h)ubeka Trust had examined 3858 claimants over the course of three years and paid out ZAR 350.3 million (approx. USD 22 million) to 2034 claimants or families of deceased claimants, with further payments pending [39]. This effort, successful as it was, will be dwarfed by that required of the Tshiamiso Trust mentioned in the Introduction, with its open-ended class of ex-gold miners and time-limited mandate of twelve years. The statutory compensation system, which has struggled to overcome decades-old backlogs, is faced with an even more extensive mandate covering all miners across all commodity types, and a wider range of lung diseases [14,40]. The high administrative costs involved in finding, examining, adjudicating, and paying ex-miners has turned attention in recent years to efficiency without comprising equity of access [14,15].

Using information from a Trust that had almost completed its work, this study found that the use of length of service (with or without latency variables) to prioritise or triage claims examinations, or to make decisions on radiologically borderline cases, results in only moderate accuracy, with potentially significant misclassification of individual claims. This caution needs to be factored into screening and clinical protocols. The study also shows that self-reported length of service shows reasonable agreement with officially recorded length of service, a finding that should inform managing claims with diagnosed silicosis in miners who cannot produce documentary evidence of mining service. 

## Figures and Tables

**Figure 1 ijerph-19-03562-f001:**
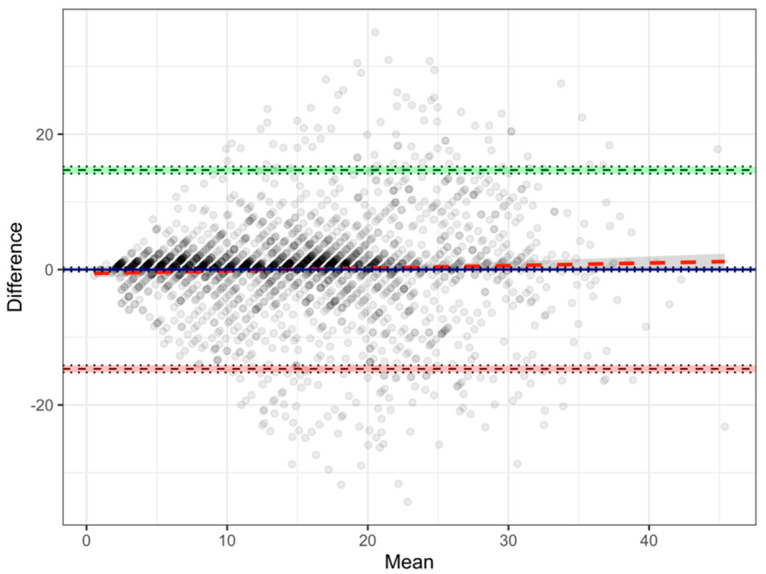
Bland-Altman plot of officially recorded vs. self-reported length of mining service (n = 2379).

**Figure 2 ijerph-19-03562-f002:**
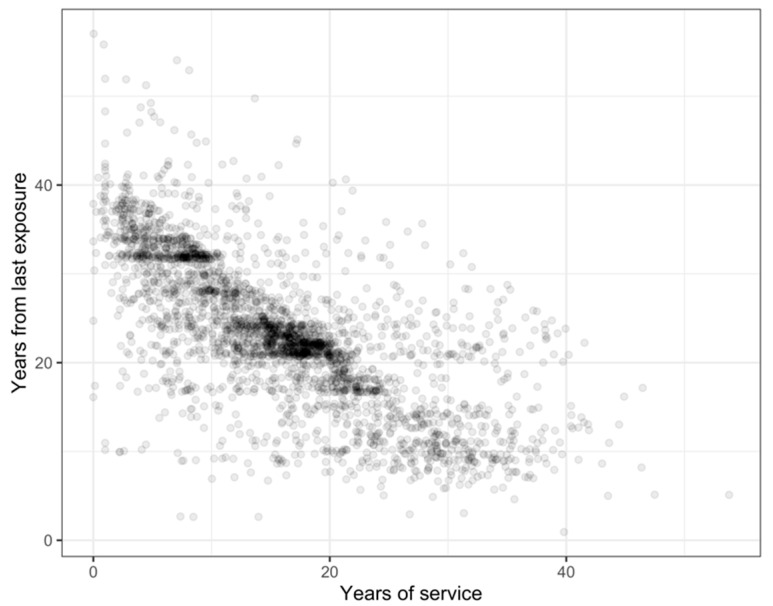
Scatterplot of length of service against time since last exposure.

**Table 1 ijerph-19-03562-t001:** Compensable and non-compensable silicosis claimants by length of service, age, and years since first and last exposure (*N* = 3146).

Category	Compensable (*N* = 1736) Count, % of Row Total		Non-Compensable (*N* = 1410) Count, % of Row Total		Row Total (*N* = 3146) Count
Years of service ^1^	*n*	%	*n*	%	*n*
0–2.9	50	32.1	106	67.9	156
3–5.9	101	31.8	217	68.2	318
6–9.9	197	39.7	299	60.3	496
10–14.9	285	51.3	271	48.7	556
15–19.9	453	63.2	264	36.8	717
20–24.9	262	68.6	120	31.4	382
25–29.9	170	74.2	59	25.8	229
30+	208	80.3	51	19.7	259
Unknown	10	30.3	23	69.7	33
Mean years	18.1 (95% CI 17.7, 18.5)		12.6 (95% CI 12.2, 13.1)		
Age (years) at claim	*n*	%	*n*	%	*n*
<50	2	9.5	19	90.5	21
50–59.9	259	45.0	317	55.0	576
60–69.9	823	55.6	657	44.4	1480
70–79.9	500	62.0	306	38.0	806
80+	152	62.8	90	37.2	242
Unknown	0	0.0	21	100.0	21
Mean years	67.5 (95% CI 67.1, 67.9)		65.3 (95% CI 64.8, 65.7)		
Years from start of service to claim	*n*	%	*n*	%	n
<20	4	20.0	16	80.0	20
20–29.9	65	46.8	74	53.2	139
30–39.9	728	51.8	678	48.2	1406
40–49.9	753	62.0	461	38.0	1214
50+	139	69.8	60	30.2	199
Unknown	47	28.0	121	72.0	168
Mean years	40.1 (95% CI 39.8, 40.4)		38.1 (95% CI 37.8, 38.5)		
Year from end of service to claim	*n*	%	*n*	%	*n*
<10	96	62.3	58	37.7	154
10–19.9	476	69.9	205	30.1	681
20–29.9	836	58.1	602	41.9	1438
30–39.9	260	40.2	386	59.8	646
40+	21	35.6	38	64.4	59
Unknown	47	28.0	121	72.0	168
Mean years	21.8 (95% CI 21.5, 22.2)		25.2 (95% CI 24.8, 25.6)		

CI, confidence interval. ^1^ Officially recorded.

**Table 2 ijerph-19-03562-t002:** Correlation between covariates with 95% confidence intervals.

	Years Worked	Years from First Exposure to Claim	Years from Last Exposure to Claim
**Age**	0.17 (0.13, 0.20)	0.53 (0.50, 0.55)	0.21 (0.18, 0.25)
**Years worked**		0.49 (0.46, 0.51)	−0.71 (−0.72, −0.69)
**Years from first exposure to claim**			0.27 (0.24, 0.30)

**Table 3 ijerph-19-03562-t003:** Unadjusted and adjusted predictive models of compensable silicosis.

Variable (5-Year Increments)	Unadjusted Odds Ratio (95% CI)	Adjusted Model A- Odds Ratio (95% CI)	Adjusted Model B- Odds Ratio (95% CI)
**Age**	1.19 (1.14, 1.25)	1.21 (1.14, 1.28)	1.21 (1.14, 1.28)
**Length of service**	1.46 (1.39, 1.52)	1.46 (1.38, 1.54)	1.32 (1.22, 1.42)
**First exposure, to claim**	1.27 (1.20, 1.35)	0.91 (0.84, 0.98)	-
**Last exposure, to claim**	0.77 (0.74, 0.81)	-	0.90 (0.84, 0.98)

CI, confidence interval.

**Table 4 ijerph-19-03562-t004:** Claim outcome of workers with <10 years of service, stratified by years since last exposure.

		Compensable		Non-Compensable	
Years from Last Exposure to Claim		n	% (95% CI)	n	% (95% CI)
**≤10**	11	1	9.1% (0.5, 37.7)	10	90.9% (62.3, 99.5)
**10.1–20**	54	29	53.7% (40.6, 66.3)	25	46.3% (33.7, 59.4)
**20.1–30**	266	86	32.3% (27.0, 38.2)	180	67.7% (61.8, 73.0)
**30.1–40**	519	202	38.9% (34.8, 43.2)	317	61.1% (56.8, 65.2)
**>40**	37	13	35.1% (21.8, 51.2)	24	64.9% (48.8, 78.2)
**Total**	887 ^1^	331	37.3% (34.2, 40.5)	556	62.7% (59.5, 65.8)

^1^ Excluding those with missing service records.

**Table 5 ijerph-19-03562-t005:** Area under the curve (AUC) of different predictive models.

Predictor	AUC (95% CI)	Nagelkerke R^2^
Unadjusted years of service	0.681 (0.662, 0.700)	0.122
Unadjusted age	0.583 (0.563, 0.603)	0.026
Unadjusted years from first exposure	0.589 (0.568, 0.609)	0.030
Unadjusted years from last exposure	0.625 (0.605, 0.646)	0.055
Adjusted model A	0.690 (0.671, 0.709)	0.135
Adjusted model B	0.690 (0.671, 0.709)	0.135

**Table 6 ijerph-19-03562-t006:** Length of service threshold, sensitivity, and specificity based on receiver operating characteristic curve for compensable silicosis (N = 3146).

	Threshold	Sensitivity	Specificity
Youden’s Index	15.7 years	59.8%	68.3%
**Fixed sensitivity**			
90%	6.5 years	90%	27.1%
80%	9.9 years	80%	44.6%
70%	13.1 years	70%	57.2%
**Fixed specificity**			
90%	23.5 years	25.3%	90%
80%	18.8 years	42.5%	80%
70%	16.1 years	57.8%	70%

## Data Availability

The data used in this study are not available in the public domain.

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
