# Peer review of "The Utility of Length of Mining Service and Latency in Predicting Silicosis among Claimants to a Compensation Trust"

_ijerph, 2022, doi:10.3390/ijerph19063562_

Round 1

Reviewer 1 Report

Dear Authors, 

I have read your manuscript with interest. In my opinion, it is a well conducted study on a large database of miners and its mean limitation is the fact that occupation exposure was based on length without including type of employment thus precluding the knowledge of the intensity of exposure. 

Specific comments

Was type of employment available from the questionnaires?

I understand that self-reported information was the one reported by the claimant and already registered in the medical history by the doctor. was this information systematically asked by the doctor? Otherwise, I would like to know the percentage of missing of these data.

695 claimants did not have chest x-rays available and were excluded. All of them had died. Were death causes available? The probability of selection bias should be discussed since the exclusion of claimants with higher percentage of silicosis would have underscored the correlations.

165 claimants with ILO score<0/1 were re-readed and were finally compensated. Authors should explain the need of double reading in all claimants. 

In Table 1 the percentage of compensable workers was maximal with 15-20 years of service, while longer services has lower percentages. I guess that intensity of exposure according to different employments could explain this disparity. The moderate accuracy of the metrics in Table 6 could also be due to the lack of a trustable measurement of the exposure.

Self-reported exposure had a reasonable agreement with recorded exposure. I think authors should discuss more extensively the chance to compensate workers lacking recorded exposure times based exclusively on self-reported information.  

Author Response

Reviewer comments

Response

Was type of employment available from the questionnaires?

Not from the questionnaires, but was available from the dataset. Job title was extremely heterogeneous as the different mines had different names for the same job. This has been expanded on in the methods and discussion sections.

I understand that self-reported information was the one reported by the claimant and already registered in the medical history by the doctor. was this information systematically asked by the doctor? Otherwise, I would like to know the percentage of missing of these data.

Yes, years of service was included in the medical history questionnaire and asked by the doctor however there was missing data. There was no information on 33 claimants (1%)

The section under “Calculated vs self-reported length of service” describes the breakdown of missing data.

695 claimants did not have chest x-rays available and were excluded. All of them had died. Were death causes available? The probability of selection bias should be discussed since the exclusion of claimants with higher percentage of silicosis would have underscored the correlations.

Unfortunately cause of death was not available. It is possible that longer service workers with more severe silicosis were excluded by death. This potential bias has been added to the limitations

165 claimants with ILO score<0/1 were re-readed and were finally compensated. Authors should explain the need of double reading in all claimants. 

Claims were reread either automatically as part of the Trust protocols (e.g. readings of 0/1 were read by another panel) or through an appeals process. This is detailed in the 3rd paragraph of the methods.

In Table 1 the percentage of compensable workers was maximal with 15-20 years of service, while longer services has lower percentages. I guess that intensity of exposure according to different employments could explain this disparity. The moderate accuracy of the metrics in Table 6 could also be due to the lack of a trustable measurement of the exposure.

The likelihood of a diagnosis of silicosis increased with longer service:

Of the 717 miners with service 15-19.9 years 453 (63%) were compensable.

Of the 380 miners with service 20 -24.9 years 262 (68,9% )were compensable

Of the 229 miners with service 25-29.9 years, 170 (74%) were compensable

And of the 259 miners with service `>30 years 208 (80%) were compensable

Table 1 has been updated to reflect row totals instead of column totals in order to reflect this stepwise increase.

Self-reported exposure had a reasonable agreement with recorded exposure. I think authors should discuss more extensively the chance to compensate workers lacking recorded exposure times based exclusively on self-reported information.  

The discussion on this point has been expanded.

Reviewer 2 Report

The paper is on “The utility of length of mining service and latency in predict-2 ing silicosis among claimants to a compensation trust”.

How does the author prove this study about the utility of mining services?

The compensation trust should be explained in detail.

What is the authenticity of the data? There is no mention of the data source. If it is taken from the existing literature, please make a comparison table to compare those studies.

Comparative studies are needed to show the novelty of the model. Future extension and insights should be provided based on the following studies:  Synergic effect of reworking for imperfect quality items with the integration of multi-period delay-in-payment and partial backordering in global supply chains; Dynamics of Cardiovascular Muscle Using a Non-Linear Symmetric Oscillator

Author Response

Reviewer comments

Response

How does the author prove this study about the utility of mining services?

Please see the discussion paragraph titled “prediction” where we discuss the utility of using length of mining service as a means to predict a diagnosis of silicosis

The compensation trust should be explained in detail.

We explained the Trust process in the “materials and methods” section of the manuscript and have added some further detail

What is the authenticity of the data? There is no mention of the data source. If it is taken from the existing literature, please make a comparison table to compare those studies.

The data are drawn from the Q(h)ubeka Trust database as explained in the “materials and methods” section of the manuscript.

Comparative studies are needed to show the novelty of the model. Future extension and insights should be provided based on the following studies:  Synergic effect of reworking for imperfect quality items with the integration of multi-period delay-in-payment and partial backordering in global supply chains; Dynamics of Cardiovascular Muscle Using a Non-Linear Symmetric Oscillator

We could find no comparative studies dealing with the analysis of silicosis data from a compensation trust. The references seem to be to a paper on inventory management and a laboratory study of cardiovascular muscle.